# Studying Physiological Synchrony in Couple Therapy through Partial Directed Coherence: Associations with the Therapeutic Alliance and Meaning Construction

**DOI:** 10.3390/e24040517

**Published:** 2022-04-06

**Authors:** Evrinomy Avdi, Evangelos Paraskevopoulos, Christina Lagogianni, Panagiotis Kartsidis, Fotis Plaskasovitis

**Affiliations:** 1Department of Psychology, Aristotle University of Thessaloniki, 541 24 Thessaloniki, Greece; clagogia@psy.auth.gr (C.L.); fplaskas@psy.auth.gr (F.P.); 2Department of Psychology, University of Cyprus, Nicosia 1678, Cyprus; paraskevopoulos.evangelso@ucy.ac.cy; 3School of Medicine, Aristotle University of Thessaloniki, 541 24 Thessaloniki, Greece; panos.kartsidis@auth.gr

**Keywords:** physiological synchrony, heart rate, therapeutic alliance, psychotherapy process, couple therapy

## Abstract

In line with the growing recognition of the role of embodiment, affect and implicit processes in psychotherapy, several recent studies examine the role of physiological synchrony in the process and outcome of psychotherapy. This study aims to introduce Partial Directed Coherence (PDC) as a novel approach to calculating psychophysiological synchrony and examine its potential to contribute to our understanding of the therapy process. The study adopts a single-case, mixed-method design and examines physiological synchrony in one-couple therapy in relation to the therapeutic alliance and a narrative analysis of meaning construction in the sessions. Interpersonal Physiological Synchrony (IPS) was calculated, via a windowed approach, through PDC of a Heart Rate Variability-derived physiological index, which was measured in the third and penultimate sessions. Our mixed-method analysis shows that PDC quantified significant moments of IPS within and across the sessions, modeling the characteristics of interpersonal interaction as well as the effects of therapy on the interactional dynamics. The findings of this study point to the complex interplay between explicit and implicit levels of interaction and the potential contribution of including physiological synchrony in the study of interactional processes in psychotherapy.

## 1. Introduction

This study rests on the assumption that psychotherapy relies on both implicit and explicit processes and that both need to be taken into account when studying clinical process [1,2]. It focuses on one aspect of implicit interaction, interpersonal physiological synchrony (IPS), and introduces the use of Partial Directed Coherence as a metric for operationalizing IPS in psychotherapy sessions. Using a single-case, mixed-method design on one couple therapy, physiological synchrony is examined in relation to the therapeutic alliance and a qualitative analysis that draws upon narrative principles of meaning reconstruction in the sessions.

Synchrony is observed in many complex biological systems and is assumed to occur through nonlinear dynamic processes rather than simple causal links. In social interaction, synchrony concerns the temporal covariation of behavior or internal states in interacting partners and can be broadly defined as ‘the social coupling of two (or more) individuals in the here-and-now of a communication context that emerges alongside, and in addition to, their verbal exchanges’ [3] (p. 558).

A key concept in the literature on interactional synchrony is interpersonal coordination, which refers to the degree to which the behaviors of interacting partners are nonrandom, patterned or synchronized in timing and form [4]. There is ample evidence that behavioral matching and interactional synchrony are ubiquitous features of human interaction, on both verbal (e.g., vocal tone, word choice, laughter, speech accent, syntax, intonation) and nonverbal (e.g., posture, gesture, facial expression, orientation, etc.) levels. Interpersonal coordination emerges early in life and is an automatic, non-conscious process that is associated with liking, affiliation, rapport, cooperation, self–other merging, perspective taking, empathy, smoothness of interaction, prosocial behaviors, compassion and increased performance in tasks that rely on joint actions [5,6,7].

In the context of psychotherapy, ‘being in sync’ has been examined primarily in relation to nonverbal behaviors and has been shown to be associated with important psychotherapy processes, such as rapport [8], therapist empathy, the therapeutic alliance [6,7], session quality and therapy outcome [9,10,11], as well as mental state in relation to attachment [12,13,14]. Drawing upon developmental research, several authors have proposed that synchronous behaviors between therapist and client are crucial for the formation of the therapeutic alliance, which in turn promotes affect regulation in the client and fosters therapeutic change [7]. Similarly, research on infant development suggests that repeated experiences of biobehavioral synchrony between infants and their parents are central to the development of affect regulation capacities in the infant and security of attachment [15,16,17,18]. There is some evidence that synchrony is associated with affect regulation in adulthood as well, as interacting partners in close relationships coregulate their arousal around a homeostatic optimal level [19,20].

### 1.1. Interpersonal Physiological Synchrony

In addition to studying synchrony in observable behavior, in recent years, there has been a growing interest in the role of synchrony in physiological arousal in psychotherapy. This is in line with the recognition that psychological and social processes cannot be isolated from embodiment and affect [21,22]. The inclusion of affective and embodied aspects of interaction is arguably particularly relevant to psychotherapy, given that affect is intimately linked with meaning construction and forms an integral part of the work of therapy [23]. The Autonomic Nervous System (ANS) plays a key role in cognition, emotion and behavior [24], and although ANS activation is not specific to affect, most emotions are associated with increased physiological arousal [25,26]. As such, several recent studies include psychophysiological measures in psychotherapy process research and treat physiological activation, and particularly its arousal component, as an index of affect [27,28]. In this literature, it is assumed that measures of psychophysiology enable the study of aspects of the therapy process that may not be accessible through self-report or observation, and can therefore add another layer of information on clinical process [29]. In other words, psychophysiological measures may reflect non-conscious, implicit affective processes and can then be used as correlates of implicit intra- and interpersonal processes in therapy [23,30]. In addition to these theoretical developments, technological advances make the continuous recording of physiological states in therapy relatively easy and unobtrusive.

Research on interpersonal physiology concerns the temporal coregulation of physiological activation in interacting partners, using continuous measures of physiological activity. The indices of ANS arousal most commonly used include electrodermal activity (EDA), considered to reflect sympathetic arousal, and variables associated with heart rate (e.g., heart rate variability), which are associated with both sympathetic and parasympathetic activity. Due to the sufficient time resolution of these variables [31], their outcome may be used to estimate the influence that one person’s physiological indices exert over another’s, through a model of physiological interactions or coupling [32,33]. In this context, interpersonal physiological synchrony (IPS) is defined as ‘any interdependent or associated activity identified in the physiological processes of two or more individuals’ [34] (p. 2). Recent reviews of studies of IPS in different interactional contexts suggest that physiological synchrony is a robust phenomenon identifiable through different methods [11,34].

In the context of psychotherapy, physiological synchrony between therapists and clients was first examined in a series of studies in the 1950s in relation to rapport and empathy [35]. More recently, the role of IPS in the psychotherapy process has been examined in several studies of psychotherapy sessions [3,23,30,36,37,38,39,40], as well as simulated sessions [13,38,41]. In a recent review of this literature, Kleinbub [14] concluded that physiological synchrony in psychotherapy is an established fact, although its clinical meaning is far from known.

Physiological synchrony in psychotherapy has primarily been associated with empathy [11,16,42,43]. However, research in interactional contexts other than psychotherapy suggest that physiological synchrony is not uniquely associated with empathy and is not necessarily positive for interactions. For example, research on infant development [17,18] shows that attachment security is associated with medium-range synchrony in parent–infant interaction and that ‘too much’ synchrony is predictive of attachment insecurity. Similarly, findings regarding the role of physiological linkage in the quality of adult romantic relationships are mixed, with several studies showing that increased physiological linkage in couples tends to be associated with poorer relationship satisfaction and the escalation of negative affect [44]. The evidence to date suggests that, in the context of negative interactions, IPS is associated with relationship dissatisfaction and conflict, whereas in positive interactions, it is primarily associated with empathy and rapport [34]. In addition to the *affective valence* of interactions, the *degree of emotional arousal* may also moderate physiological synchrony; for example, in studies of mother–infant interactions, higher maternal heart rate, thought to reflect increased affective arousal, has been associated with lower physiological synchrony with her infant [45]. Drawing upon these findings, it seems important for future research to take into account the characteristics of the relational context when studying the role of IPS in psychotherapy.

A related issue concerns the way IPS is conceptualized, operationally defined and calculated. The majority of studies to date of IPS in psychotherapy examine only positive correlations, i.e., in-phase synchrony, where the therapist’s and client’s arousal covary in the same direction, and assume that negative correlations, or anti-phase synchrony, reflect lack of synchrony. Other studies, however, suggest that anti-phase synchrony, where one partner’s physiological arousal decreases as the other partner’s increases, reflects processes of coregulation or complementarity [46,47]. For example, in one study implicating a storytelling task, it was found that the narrator’s autonomic arousal decreased when the listener’s increased and he or she displayed affiliation; this was interpreted as reflecting a process of ‘sharing the emotional load’, whereby the listener’s engagement regulated the teller’s physiological arousal [48]. Similarly, in a study of ANS activation in psychoanalytic therapy, the therapists’ empathic displays were associated with increased arousal in the therapist and decreased arousal in the client, whereas sequences of the therapists’ challenges were associated with increases in both participants’ arousal [49]. In line with these findings, Butler & Randell [19] suggest that asynchrony may be associated with stress buffering, whereby one individual moderates the stress level of another. Based on the above, including both in-phase and anti-phase synchrony in studies of IPS in psychotherapy is likely to provide a more nuanced approach to understanding this multifaceted interactional phenomenon.

The metric employed to estimate interaction is also of importance. Most studies investigating IPS use correlation-derived estimates, which are sensitive to spurious correlations and do not address causality or directionality in the interaction [14]. In order to overcome this issue, approaches that employ specific causality tests, such as Granger causality, adjusted for estimating the information flow between multivariate time series can be used in the frequency domain [50]. Combined with surrogate testing of the parameters used to estimate interactions [11], such approaches may be combined with a windowed analysis to reach a stable and fine-grained temporal resolution that can also provide directionality.

Another important issue when examining physiological synchrony in psychotherapy relates to the timescales employed in the analysis. Most studies calculate IPS over whole sessions, despite the fact that IPS is likely to be a transient phenomenon that fluctuates through sessions [1,34]. Similarly, recent studies approach the therapeutic alliance as a dynamic phenomenon and show that therapy sessions contain several periods characterized by ruptures in the alliance, often followed by interactive repair [51]. Indeed, several authors suggest that it is precisely such repairs that are important for optimal development and therapeutic change [51,52,53]. Therefore, examining synchrony on a more micro-level of interaction can shed light on processes that may not be apparent at the session level.

In sum, research on physiological synchrony in psychotherapy suggests that it can add important information regarding the psychotherapy process; given that IPS may reflect different interactional processes—including empathy, affect coregulation and conflict—caution is needed when interpreting findings. Moreover, the field is fragmented on both conceptual and methodological levels, as reflected in the prevalent lack of agreement on terminology, data collection methods, research designs and statistical analyses [11,34,54]. Recent reviews suggest that, given how little we know about the context-specific factors that affect IPS, it may be preferable to use idiographic designs and theoretically informed analyses of the therapy process. Since the publication of these reviews, a few such studies have been published that shed light on the different functions of physiological synchrony in psychotherapy [23,30,37,42,55,56,57].

Before turning to the current study, we briefly discuss the concept of the therapeutic alliance, with a focus on couple therapy, given that it is a key clinical concept that has been associated with physiological synchrony.

### 1.2. The therapeutic Alliance in Couple Therapy

Several contemporary approaches to psychotherapy adopt a discursive and narrative perspective and conceptualize the process of change in psychotherapy in terms of meaning reconstruction [58]. In this framework, psychotherapy is described as a semantic process that relies on the creation of a dialogical space, which facilitates the reconstruction of clients’ life narratives so that they become more complex, polyphonic, emotionally salient, inclusive and flexible [59]. The therapist’s receptive and relationally responsive attitude towards the clients’ storytelling and expression of affect are considered crucial elements in this process [60]. There is ample evidence that different therapist actions associated with responsiveness play an important role for the process and outcome of psychotherapy [61], with the therapeutic alliance being a key relational aspect in this process.

The therapeutic alliance is a pan-theoretical concept that is associated with the collaborative aspects of the therapeutic relationship and has been extensively studied as an important process variable in psychotherapy. It is usually conceptualized as comprising three interlinked aspects: a strong emotional bond between clients and therapists, and agreement and collaboration on the goals and the tasks of therapy [62]. The quality of the therapeutic alliance has consistently been shown to be a predictor of outcome in individual psychotherapy across different modalities [63], as well as couple and family therapy (CFT) [64,65,66,67,68,69]

In conjoint treatments, such as couple therapy, the therapeutic alliance consists of a web of interlinked relationships between participants and the various subsystems thus formed [63,66]. Several factors—such as power dynamics, conflict, trust, loyalties and secrets in the couple or family—affect the formation of the alliance in CFT [69,70,71]. A strong overall alliance in couple therapy requires a balanced alliance between the therapist and each partner, as well as agreement in the couple on the problems, goals and values of therapy; as such, the therapist is encouraged to foster an alliance with each partner, avoiding ‘split alliances’, and to promote within-couple alliance [66].

The current study is a mixed-method, single-case study aiming to illustrate the potential of the PDC metric as a useful way of examining IPS in relation to the therapy process; it assumes a theoretically driven idiographic design and examines whether the therapeutic alliance maps onto IPS findings.

## 2. Materials and Methods

The research material in this study is drawn from one-couple therapy, conducted in an outpatient Family Therapy Department in Greece, in the context of a wider naturalistic, multisite research study [30,39,72]. The treatment in this service follows systemic principles and includes the use of reflective conversations with a co-therapist. In usual clinical practice, sessions are provided monthly; a second therapist watches the session between the primary therapist and the couple behind a one-way mirror and joins them for a reflective conversation towards the end of each session [73]. Participating couples were informed about the study by a graduate researcher at the end of their first session. Participation in the project was voluntary, and ethical approval was granted by the Family Therapy Department’s Scientific Board. Both clients and therapists gave permission for the data to be used for research purposes.

### 2.1. The Case

This therapy consisted of 15 sessions spanning 14 months. The couple, Costas and Demetra, is a white heterosexual couple in their mid-thirties. Demetra is a law graduate with a successful professional career. Costas has no university education; he worked as a technician in the past and is currently unemployed. The couple had been in a long-term relationship of over 10 years when they came to therapy. They sought therapy because of increasing tension in their relationship following the birth of their baby 10 months earlier. Two experienced female clinical psychologists and systemic family therapists in their fifties participated in this therapy. The therapy centered on Demetra’s distress in her role as a mother, the expression of anger and conflict between the spouses, and Costas’ low self-esteem associated with periods of unemployment. At the end of treatment, the couple reported an improvement in their personal lives and their relationship.

### 2.2. Procedure

All sessions were video-recorded in split-screen mode with four web-cameras. In addition, in two sessions (sessions 3 and 14), physiological measures of the participants’ heart rates were recorded for the duration of the session. Within 24 h of the measurement sessions, a graduate researcher conducted separate Stimulated Recall interviews [74,75] with each client and therapist, each lasting approximately 30 min.

### 2.3. Measures

#### 2.3.1. Autonomic Nervous System Responses

The participants’ autonomic nervous system (ANS) responses were recorded via Firstbeat Bodyguard (Firstbeat Technologies, Jyväskylä, Finland) [76] mobile heart rate (HR) monitors. Ag/AgCl electrodes, connected to the Firstbeat Bodyguard, were attached on two sites on the skin of the chest before the start of each measurement session and were removed the next day, with the guidance of a graduate researcher. HR was continuously recorded during this period.

#### 2.3.2. Clinical Outcomes in Routine Evaluation–Outcome Measure (CORE-OM)

The outcome of therapy was examined using the CORE-OM, administered at the start and end of therapy. The CORE-OM is a widely used, 34-item self-report measure that examines psychological distress in four domains: wellbeing, problems, functioning and risk [77].

#### 2.3.3. Session Rating Scale (SRS)

The SRS is a four-item, ultra-brief visual analogue instrument to assess the global strength of the alliance, designed to be used in routine outcome monitoring [78]. The four items measure the therapist–client emotional bond, agreement on goals, agreement on tasks and overall rating of the alliance. It is scored by summing the marks measured to the nearest centimeter on each of the four lines. Based on a total possible score of 40, any score lower than 36 overall, or 9 on any scale, could be a source of concern.

#### 2.3.4. System for Observing Family Therapy Alliances (SOFTA-o)

The SOFTA-o is an observer-based measure developed to study the therapeutic alliance in couple and family therapy [79]. It examines the contribution of each participant to the alliance by coding specific behaviors in four dimensions: Emotional connection, Engagement in the therapeutic process, Safety within the therapeutic system and Shared sense of purpose. The first three dimensions concern the therapist(s)–clients relationship, whereas the fourth concerns the couple sub-system. Following the coding of specific items, global ratings are provided for each dimension on a 7-point ordinal scale, ranging from -3 (extremely problematic) to +3 (extremely strong), with 0 denoting an unremarkable or neutral alliance. These dimensions are conceptually interdependent and moderately correlated and can be combined in a composite score [66].

### 2.4. Data Analysis

#### Interpersonal Physiological Synchrony

Data from the ANS were analyzed using Firstbeat PRO Wellness Analysis Software^®^ version 1.4.1. This software uses neural network modeling to calculate Heart Rate Variability (HRV) indices second-by-second. This is achieved using a short-time Fourier Transform method (STFT) combining data from HR- and HRV-derived variables that describe respiration rate and oxygen consumption (VO2). In addition, the absolute stress vector (ASV) is calculated from the HR, high-frequency power (HFP), low-frequency power (LFP) and HRV-derived respiratory variables, as an index of the activity of the sympathetic nervous system. The ASV grounds on detecting sympathetic reactivity that exceeds the momentary metabolic requirements of the ANS. Hence, the ASV is high when the heart rate is elevated, HRV is low and respiration rate is low relative to HR and HRV [80]. The ASV is calculated at a 1 Hz rate.

### 2.5. Partial Directed Coherence within Sessions

Within-session, directed, interpersonal physiological synchrony based on ASV was estimated using Partial Directed Coherence (PDC) [50]. PDC analysis transforms the ASV time series into the frequency domain and provides time-lagged associations between two participants’ multivariate signals, assessing their statistical independence or predictability [50]. Specifically, grounded on instantaneous Granger causality, it implies that, knowing the previous states of the first signal (the leading signal), one may achieve a better prediction of the second signal (the pacing signal), than just knowing the previous states of the second signal. Hence, it describes the direction of information flow between isolated pairs of time series, in a frequency-domain representation of the notion of Granger causality. This approach has recently been proposed as a method of choice for estimating IPS in psychotherapy by Kleinbub [54], due to its ability to establish direction, and thus causality, in interactions. Due to the time-varying conditional variance of HRV signals [81], PDC as a frequency-domain method for identifying causal interactions between the signals was preferred over the classical Granger causality, which estimates interactions in the time domain. In addition, PDC has previously been used to successfully estimate the frequency-domain causality in cardiovascular time series with Instantaneous Interactions [82].

The second-by-second ASV data of the measurement sessions were imported into Matlab (MathWorks Inc., Natick, MA, USA) as time series. The ASV time series were segregated into time-windows of 50 s, and the PDC for each window was estimated independently for each pair of participants in each session via an in-house script based on the work of Baccalá and Sameshima [50]. The length of the time-window was empirically determined on the basis of a series of tests comparing the number of significant PDC time-windows within independent sets of surrogate data generated via Matlab, aiming to achieve the best possible balance between the resolution of the analysis (i.e., smallest time-window) and the absence of false-positive significant PDC time-windows. Hence, for each 50 s time-window of the session, we retrieved two PDC values for each pair of participants (one for each direction, i.e., one in which participant 1 leads and participant 2 paces, and one in which participant 2 leads and participant 1 paces). Additionally, a statistical test based on Monte Carlo iterations of the corresponding data was performed for each pair, in order to identify time-windows with a significant PDC. The threshold of significance was defined as *p* = 0.05/3, accounting for the total number of comparisons in which the same set of data participated, thereby effectively controlling for multiple comparisons. Only significant PDC values were taken into account.

### 2.6. Partial Directed Coherence between Sessions

The number of significant PDC time-windows for each pair of participants was compared between sessions 3 and 14 as an index of the overall effect of therapy on interpersonal physiological synchronization. The aim was to identify differences in the global characteristics of IPS between sessions at the start and end of therapy.

### 2.7. Qualitative Analysis of the Therapy Process

#### 2.7.1. Topical Episodes

Τhe measurement sessions were segmented into topical episodes, i.e., periods of time during which a specific topic was discussed [83]. This coding was initially carried out by two graduate researchers and was checked by third researcher, and any discrepancies were resolved through discussion. This initial thematic coding provides a description of the main themes discussed in a session. Session 3 was segmented into 14 topical episodes, ranging from 2 to 15 min’ duration, and session 14 was segmented into 12 topical episodes, ranging from approximately 1 to 9 min’ duration.

#### 2.7.2. Therapeutic Alliance

Two graduate psychologists, trained in using the SOFTA-o, coded each session. The raters coded the sessions independently and then discussed any discrepancies until consensus was reached. Next, in order to gain a more fine-grained coding of the development of the alliance through the session, the strength of the alliance was coded for each topical episode.

## 3. Findings and Discussion

With regards to the outcome of therapy, the clients’ CORE-OM scores decreased significantly over the course of therapy, suggesting a clinically significant reduction in psychological distress (Table 1). At the onset of therapy, both partners reported a medium level of distress, and, importantly, Costas scored on items concerning the risk of harming himself. At the end of therapy, Demetra’s CORE-OM score decreased to the cut-off point for clinical distress (<10), and Costas’ showed clinically significant change (>5 clinical score points) [77]. In terms of the therapeutic alliance, Costas’ scores indicated a positive alliance in session 3, which further increased in the penultimate session, whereas Demetra’s scores indicated a problematic alliance in session 3, which improved in the penultimate session.

Next, we present the key quantitative findings regarding interpersonal physiological synchrony (IPS) within and across the two measurement sessions. Then, the potential of PDC analysis as a useful way of examining the process of therapy is explored through a mixed-method analysis of session 3.

The physiological activity of the couple, as reflected in their ASV, in the two sessions is presented in Figure 1. In both sessions, Demetra’s autonomic arousal decreased as the session progressed, whereas Costas’ remained relatively constant through. It is worth noting that Demetra’s mean ASV score in the penultimate session was significantly higher than in the third session, and her arousal shows higher variance. The clinical relevance of this observation would require further investigation and lies beyond the scope of this study.

### 3.1. Interpersonal Physiological Synchrony

#### 3.1.1. IPS in Session 3

The PDC analysis identified 29 time-windows in which the participants’ ASV were synchronized in session 3, out of a total of 93 time-windows (Table 2 and Figure 2). This corresponds to at least two participants’ physiological arousal being synchronized in 31,2% of the total session time. More specifically, Demetra’s ASV values led Costa’s ASV in one time-window, and the therapist’s ASV in four. In contrast, Costa’s ASV led Demetra’s ASV in eight time-windows, and the therapist’s ASV in nine. Lastly, the therapist’s ASV led Demetra’s ASV in six time-windows, and Costas’ in eight. Overall, in session 3, Costas’ autonomic arousal was found to lead IPS to a greater degree than Demetra’s; moreover, the therapist had a leading role in several parts of the session, while Demetra primarily had a pacing role.

In addition, in session 3, several time-windows showed increased IPS; we use this term to describe time-windows in which more than one of the six possible directed synchronizations were observed. We consider these time-windows as particularly significant. Specifically, four time-windows showed increased physiological synchrony. Notably, in one time-window, all three participants were physiologically synchronized, with a mutual IPS between the clients and both clients’ arousal also leading the therapist’s ASV. Moreover, as can be seen in Figure 2, the time-windows with IPS tended to cluster around specific points in the session. We consider this clustering of IPS as reflecting time periods in the session that are significant for the process of therapy.

#### 3.1.2. IPS in Session 14

The PDC analysis of the penultimate session identified 10, out of a total of 58, time-windows in which the participants’ ANS arousal was synchronized (Table 3 and Figure 3). This corresponds to at least two participants’ physiological arousal being synchronized in 17.2% of the total session time. More specifically, Demetra’s arousal led Costas’ ASV in four time-windows, and the therapist’s ASV in one. Costas’ arousal led Demetra’s ASV in one time-window, and the therapist’s in two. Lastly, the therapist’s ASV led Demetra’s ASV in four time-windows, and Costas’ in one. Overall, IPS in the penultimate session was equally led by Demetra and the therapist, and both clients had similar pacing roles, with Demetra pacing the therapist’s ASV and Costas pacing Demetra’s. Three time-windows showed increased physiological synchrony in this session, and again, time-windows with PDC tended to cluster together.

#### 3.1.3. IPS between Sessions

A comparison of the PDC values between the two measurement sessions shows that (i) the total time spent in interpersonal physiological synchrony was significantly lower in session 14 as compared to session 3, and (ii) the global architecture of the interpersonal physiological synchrony network was reorganized to become more balanced as therapy progressed (Figure 4). As mentioned above, the percentage of the total session time with IPS decreased from 31.2% in session 3 to 17.2% in session 14. The IPS between the therapist and each of the clients showed the most marked decrease, from twenty-seven time-windows (28.1% of the session time) in session 3 to eight time-windows (13.7% of total session time) in session 14. This reduction in IPS over the course of therapy can be seen to reflect the clients’ reduced affective arousal and their gradual disengagement from the process. As therapy progressed, the clients’ difficult feelings and conflicts were expressed, elaborated upon and gradually reconstructed, and the physiological synchrony between the clients and the therapist decreased. This is in line with the characteristics of the closing stages of therapy, which entail less affectively charged and more reflective conversations, as well as a process of gradual disengagement from the therapeutic relationship and the work of therapy.

Furthermore, IPS in a multi-actor setting such as couple therapy is more complex, as there are six possible pairs of participants. A shift was observed in how the IPS was distributed between participants; in session 3, the IPS was mainly driven by Costas, who led Demetra’s and the therapist’s ASV in seventeen time-windows and paced the therapist in eight. In contrast, in session 14, the overall synchrony was more equally driven by all participants, producing a more balanced or ‘democratic’ network structure (Figure 4). This finding points to the co-creation of a more equally distributed and balanced embodied relatedness between participants as therapy was reaching termination.

### 3.2. Physiological Synchrony and Clinical Process

In order to deepen our understanding of the relational meaning of IPS as it fluctuated through a session, the clinical process in session 3 was qualitatively analyzed drawing upon narrative principles, and the findings were subsequently examined in relation to the PDC analysis. In brief, the session was segmented into topical episodes [83]; this thematic coding allows researchers to identify key themes in a session and track the process of meaning co-construction, thus obtaining a relatively fine-grained description of meaning making through the session. Next, a qualitative analysis was performed to identify the significant moments in the session, which were defined as those topical episodes where: (a) important issues in the couple’s life were introduced and narratively elaborated; (b) associated emotions were recognized, explored and expressed; and (c) the meaning of these key issues began to be reconstructed. The central theme in this session concerned Demetra’s low mood and her strong ambivalent feelings regarding her role as a mother. Two topical episodes were identified through the qualitative analysis as entailing the elaboration of this theme, accompanied by intense emotional expression; these are briefly described below.

The theme of Demetra’s conflicts in her role as a mother was first introduced in TE4, approximately ten min into the session (duration 10′40″). This episode started with the therapist asking how the couple would choose to spend their time together if they had the finances and caretaking support. Costas made several suggestions that Demetra firmly rejected as she felt ‘bored’ with everything. Through the therapist’s gentle curiosity and empathic questioning, Demetra’s boredom was gradually reconstructed as entailing intense sadness; Demetra cried as she described her low mood, exhaustion, and sense of suffocation in her role as a mother. Towards the end of the episode, Costas gently talked about Demetra’s lack of interest in sex, and this led to the expression of more sadness by Demetra. This episode contained the elaboration of the key theme of the session along with nonverbal displays of affect, as well as several markers of a moderately strong therapeutic alliance; this was particularly evident in the relationship between Demetra and the therapist, as well as within the couple (Table 4). The PDC analysis for this episode shows a cluster of five time-windows with IPS, accounting for 39% of the episode time; two of these time-windows show increased IPS, whereby more than two participants are in-sync (Figure 3). In other words, the findings from the PDC analysis concur with those of the narrative analysis and with the coding of the therapeutic alliance, suggesting that this topical episode was important for the therapy process on both semantic and embodied levels.

The same theme was further elaborated with increased emotional expression in TE7. This long episode (duration 14′40″) took place in the middle of the session. It started with Demetra crying as she described feeling trapped and suffocating in her relationship with their baby; she vividly described her frustration and rage towards their baby, the wish to hurt him that she sometimes experienced, her angry outbursts towards him, and the intense guilt that she felt after such outbursts. As she listened to Demetra’s emotional narrative, the therapist displayed many nonverbal signs of affiliation and empathy. She also introduced the hypothesis that Demetra’s difficulties and sense of failure result from comparing herself to an idealized version of motherhood; this was followed by a productive conversation that challenged Demetra’s idealization of her own mother, as well as the dominant discourse of ideal motherhood [60]. In terms of the alliance, this episode contained markers of a moderately strong alliance between Demetra and the therapist, whereas Costas displayed some markers of difficulty in the alliance. Based on the PDC analysis, there were five time-windows with IPS in this episode (including two windows with increased IPS), and these account for 28% of the episode time. Notably, this topical episode contained one time-window during which all three participants were physiologically synchronized. This corresponds to the point in the session where Demetra cried as she talked about the rage and guilt she felt towards their baby. A cluster of time-windows with IPS can be seen at the end of the episode, as Demetra’s sadness and sense of suffocation in her maternal role were expressed. Once again, in this topical episode, the PDC analysis identified a period in the session that entailed intense affective expression by Demetra, empathic responsiveness by the therapist and a strong therapeutic alliance.

Next, we examined a topical episode identified as significant through the PDC, but not through the qualitative analysis: TE12 contained a cluster of time-windows with IPS that account for 66% of the episode time. The episode took place towards the end of the session (duration 5′20″) and focused on Demetra’s lack of sexual desire, a delicate and affectively charged issue in the couple’s relationship. In this instance, the PDC analysis identified a part of the session that was not identified as important from a qualitative perspective, but which proved to be significant later in the treatment.

In sum, the two topical episodes that were identified as important for the process of therapy through the qualitative analysis also entailed increased IPS and increased ratings of the therapeutic alliance, as compared to the rest of the session. These findings are in line with the literature that points to the role of IPS in empathy, affiliation, rapport and the therapeutic alliance [7,11,43]. At the same time, PDC analysis proved useful in identifying significant moments in the session.

In order to explore the relational significance of IPS, we examine the findings from the PDC analysis in relation to the interactions between participants in session 3. With regards to the therapist–client(s) interaction, periods of physiological synchrony between the therapist and Demetra account for 10.1% of the session time (corresponding to 27.8% of the total time with IPS); IPS between the therapist and Costas is significantly higher, accounting for 18.3% of the total session time (corresponding to 47% of the total time with IPS). This points to the presence of a more intense affective connection between Costas and the therapist, and this is in line with the clients’ respective SRS scores (Table 1). This finding reflects the complex interplay between explicit and implicit aspects of interaction in psychotherapy. More specifically, the therapist was very responsive to Demetra’s distress on an explicit level, as she openly expressed empathy and affiliation with Demetra’s painful conflicts regarding motherhood. At the same time, she was significantly more in-sync with Costas on an embodied level. This is in line with findings that behavioral and physiological synchrony seem to be independent processes that do not always co-occur [84]. The therapist was able to maintain a balanced therapeutic alliance with both members of the couple, and this was achieved through different modalities. In other words, implicit and explicit modalities of interaction were used to manage different interactional goals [30]. This is important, as split alliances, i.e., situations where the therapist takes sides by colluding with one partner, have a negative impact on the outcome of therapy [79]. In addition, there is some evidence that the therapeutic alliance with the male partner is critical for therapy outcome in heterosexual couples [70].

With regards to the couple’s relationship, Costas’ physiological arousal led Demetra’s ASV significantly more than the opposite. Thus, although Demetra’s difficulties were central on the level of talk, on an implicit embodied level, Costas had a more powerful influence on the interaction. Again, this is an observation that illustrates the complex interplay between the verbal and embodied aspects of psychotherapeutic interaction. A possible interpretation of this observation is that of affect co-regulation, where Costas’ presence could be seen to regulate Demetra’s affective arousal, as often happens in complementary, i.e., homeostatic, couple relationships [36]. A closer examination of the level of both partners’ arousal, the valence of their affective displays, and an examination of the talk in these episodes would help contextualize and interpret these observations more fully.

## 4. Conclusions

The findings of this study point to the complex interplay between explicit and implicit levels of interaction and the potential added value of including physiological synchrony in the study of interactional processes in couple therapy [36,55,72,85,86]. In line with contemporary theories of therapeutic change, a key assumption of this work is that psychotherapy entails processes of intersubjective meaning making that take place through different modalities and, presumably, with different degrees of conscious awareness [23]. From this perspective, including measures of physiological activation in the study of psychotherapy sessions can help examine implicit, embodied interactional processes that contribute significantly to the formation of the therapeutic alliance, the co-creation of new meanings and, ultimately, therapeutic change. Although several research methods have been developed to study the talk in interaction [48,58], these methods generally fail to grasp the implicit, procedural level of interaction. Our attempt to include measures of autonomic arousal in studies of the therapy process and to operationalize implicit interactional processes of embodied responsiveness are in the spirit of exploring ways to include the implicit realm when studying the psychotherapy process [23].

Research to date suggests that IPS reflects different interactional processes, and these need to be disentangled for the field to progress [11,34,54]. In our study, we propose the use of a windowed Partial Directed Coherence-based approach as a metric to calculate physiological synchrony, as this allows a more nuanced examination of the dynamic nature of IPS in psychotherapy sessions. PDC analysis allows us to examine the therapy process in specific interactional events in the session, and this *micro-focus* provides a more fine-grained description of interactional dynamics as they develop, thus allowing a more nuanced interpretation of the role of IPS in the therapy process. Importantly, PDC analysis allows us to examine the *directionality* of synchronous interactions, which again adds another layer of complexity to our understanding of the role of physiological synchrony in the therapy process. Therefore, the proposed approach models the couple’s interactions within the setting of a therapy as a self-organizing system, a system that is both open and complex, exchanging energy and information between its component parts and with its surroundings [87]. This exchange may be synchronic and diachronic, in spatial distribution and time transitions, therefore demanding multidimensional theoretical models to represent its hybrid nature [88].

A key aim of this study was to explore the links between a quantitative approach to the study of IPS and the characteristics of interactional dynamics and the clinical process, and this mixed-method analysis produced promising initial findings. More specifically, it examined shifts in IPS between the start and end of therapy in a successful couple therapy and identified a reduction in IPS as therapy progressed. This decrease primarily concerned the therapist–client(s) interaction and was interpreted as a reflection of progress, in the sense of a decrease in the intensity of negative affects expressed by the clients and the need for therapist empathy, as well as the couple’s gradual disengagement from the process of therapy in line with the termination phase. In addition, the network of IPS between the three participants became more balanced. Both these findings are in line with a good therapy outcome, and as such, they provide support for the clinical validity of PDC analysis.

The main limitation, inherent in this approach, is that only one couple is included; hence, the descriptive outcome of the study cannot be generalized. Nonetheless, we propose that this detailed analysis provides a necessary step for evaluating the usefulness of employing PDC analysis to examine IPS in therapy sessions, which can now be further elaborated. Studying IPS via a windowed PDC approach may lead to an even more detailed identification of the underlying processes if the characteristics of the ANS signals during significant time-windows are further investigated. In addition, calculating positive vs. negative correlations of ANS activity or specific patterns of ANS reactivity within the significant time-windows may be used in future studies to examine their associations with different intersubjective processes, such as empathy, alliance or affect contagion. We hope that future work in this field will exploit the strengths of the PDC analysis and further our understanding of the embodied, relational aspects of the therapy process.

## Figures and Tables

**Figure 1 entropy-24-00517-f001:**
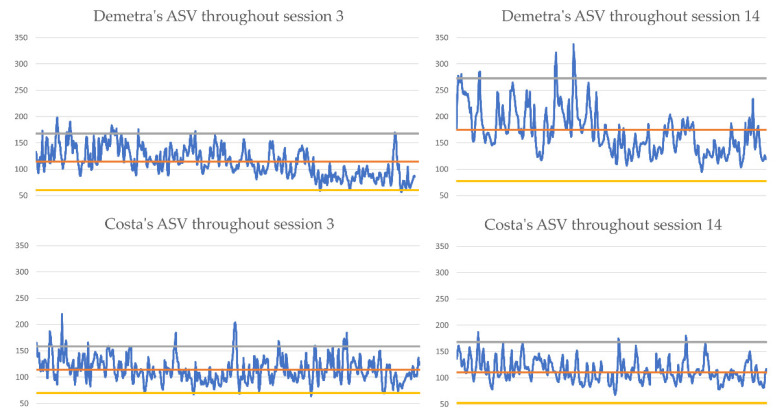
Demetra and Costa’s ASV values in the sessions, plotted against the mean ASV value for the session (red line), 2 Standard Deviations below (yellow line) and 2 SD above (grey line) the mean.

**Figure 2 entropy-24-00517-f002:**
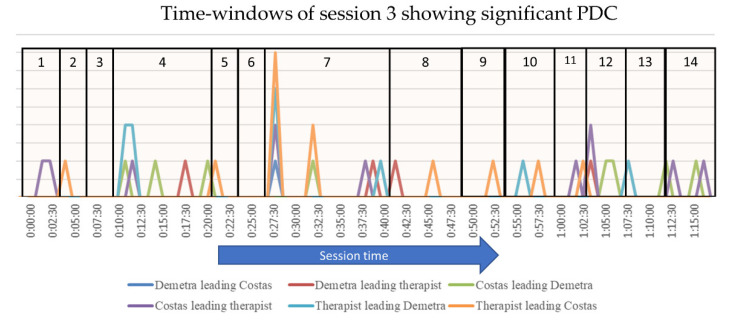
Time-windows showing significant PDC in session 3. The height of each line indicates the number of pairs that show significant PDC within each time-window. The vertical lines denote the boundaries of the Topical Episodes.

**Figure 3 entropy-24-00517-f003:**
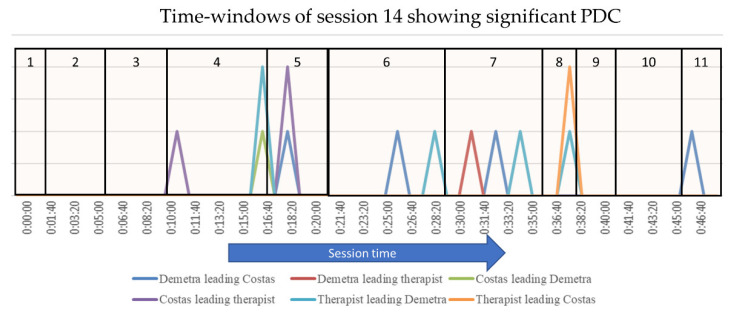
Time-windows showing significant PDC in the penultimate session (session 14). The height of each line indicates the number of pairs with significant PDC within each time-window. The vertical lines denote the boundaries of the Topical Episode.

**Figure 4 entropy-24-00517-f004:**
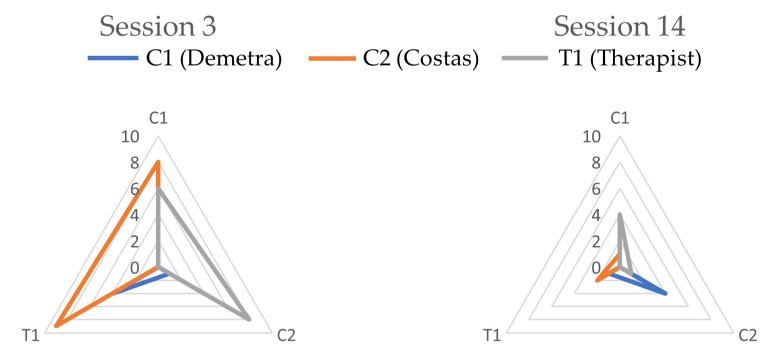
Time-windows showing significant PDC per leading participant.

**Table 1 entropy-24-00517-t001:** Clients’ CORE-OM and SRS scores.

	CORE-OM		CORE-OM RISK		SRS	
	Session 1	Session 15	Session 1	Session 15	Session 3	Session 14
Demetra	12	10	0	1.6	5.6	8.0
Costas	19	11	5	0	8.9	9.8

**Table 2 entropy-24-00517-t002:** Number of time-windows showing significant PDC synchronization between clients and therapist in session 3.

		Leading Role		
		Demetra	Costas	Therapist
Pacing role	Demetra		8	6
	Costas	1		8
	Therapist	4	9	

**Note**: Number of time-windows in session = 93. Time-windows in which at least two participants show significant PDC = 29.

**Table 3 entropy-24-00517-t003:** Number of time-windows showing significant PDC synchronization between clients and therapist in the penultimate session.

		Leading Role		
		Demetra	Costas	Therapist
Pacing role	Demetra		1	4
	Costas	4		1
	Therapist	1	2	

**Note:** Number of time-windows in session = 58. Time-windows in which at least two participants show significant PDC = 10.

**Table 4 entropy-24-00517-t004:** Comparison of SOFTA scores and number of PDC identified significant time-windows for each Topical Episode.

TE	SOFTA Scores					Time-Windows with PDC	% Episode Time in PDC
	Costas	Demetra	Therapist	SSP	Composite Score		
1	2	1	0	0	3	2	47.6
2	1	1	2	0	4	1	26.3
3	−1	0	1	2	3-1	0	0
4	1	2	1	2	6	5	39
5	0	0	0	1	1	1	15.4
6	1	0	0	0	1	0	0
7	−1	2	2	0	4-1	5	28
8	0	1	1	2	4	2	19.2
9	1	0	1	2	4	1	17.9
10	0	1	1	1	3	2	28.6
11	0	0	0	1	1	1	31.2
12	0	1	1	0	2	5	66
13	1	2	1	2	6	1	34.5
14	1	0	1	1	3	0	0

## Data Availability

The ANS data presented in this study are available on request from the corresponding author, and the data from the video material are not publicly available due to ethical restrictions.

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
