# Peer review of "Studying Physiological Synchrony in Couple Therapy through Partial Directed Coherence: Associations with the Therapeutic Alliance and Meaning Construction"

_entropy, 2022, doi:10.3390/e24040517_

Round 1
Reviewer 1 Report
-
Author Response
We have not received any comments to respond to. Thank you for your time reading our manuscript.
Reviewer 2 Report
The manuscript introduces partial directed coherence (PDC) for analyzing synchrony in single couple therapy sessions. PDC allows the determination of direction of the synchrony. In addition, the sessions are divided in topical episodes which allows the following of the therapy process at a fine temporal level. The study also shows how PDC can be used for studying the therapeutic alliance. All the methods are well described and applied. Furthermore, the introduction and conclusions are in line with the study design and results.
I would like to present some minor details below.
Is there a specific reason to use couples therapy instead of the more common couple therapy (in the title and within the text)?
Should the asterisk referring to the corresponding author be associated with the first (Avdi), instead of the last author?
Physiological (autonomic) arousal on line 69 could be replaced by physiological arousal since autonomic nervous system is introduced only after a few sentences.
Should ‘is defined a’ on line 95 be replaced with ‘is defined as’?
The threshold of significance was defined as p=0.05/3 accounting for the total number of comparisons in which the same set of data participated, thereby effectively controlling for multiple comparisons (lines 331 and 332). Number 3 obviously refers to the number of PDC coefficients calculated for each time windows. Since the analysis is directional and synchrony was analyzed for both directions, I would take this in account and correct the p-value by dividing it by 6.
Since Demetra’s session 15 CORE-OM was 10 it was not <10 as written in the text, and if the threshold for clinical significance is 10, then her score did not decrease to the non-clinical level. Right?
For better readability, please, increase the font of the title in figure 1 and possibly in figures 2 and 3 too.
While in other windows the names of the clients are shown, in figure 4 clients are referred as C1 and C2 and therapist as T1. I would like to suggest that the line captures would be changed to include the names, e.g. C1 (Demetra). Then C1, C2 and T1 can be retained in the two panels.
Since the title of figure 4 is exactly the same as the figure caption, the title may not be necessary.
In references the titles of some references have sentence cases and some have each word capitalized. Please, check if the journal will accept this.
References 77 and 80 are not accurate enough.
Author Response
Response to Reviewer 2
The manuscript introduces partial directed coherence (PDC) for analyzing synchrony in single couple therapy sessions. PDC allows the determination of direction of the synchrony. In addition, the sessions are divided in topical episodes which allows the following of the therapy process at a fine temporal level. The study also shows how PDC can be used for studying the therapeutic alliance. All the methods are well described and applied. Furthermore, the introduction and conclusions are in line with the study design and results.
I would like to present some minor details below.
Is there a specific reason to use couples therapy instead of the more common couple therapy (in the title and within the text)?
Although both terms are used in the literature, we agree that ‘couple therapy’ is more commonly used and have amended this in the title and text.
Should the asterisk referring to the corresponding author be associated with the first (Avdi), instead of the last author?
Thank you for this correction; this was an oversight on our part, and it has now been revised.
Physiological (autonomic) arousal on line 69 could be replaced by physiological arousal since autonomic nervous system is introduced only after a few sentences.
This has now been amended.
Should ‘is defined a’ on line 95 be replaced with ‘is defined as’?
Thank you for this correction; this was an oversight on our part, and it has now been amended.
The threshold of significance was defined as p=0.05/3 accounting for the total number of comparisons in which the same set of data participated, thereby effectively controlling for multiple comparisons (lines 331 and 332). Number 3 obviously refers to the number of PDC coefficients calculated for each time windows. Since the analysis is directional and synchrony was analyzed for both directions, I would take this in account and correct the p-value by dividing it by 6.
The reviewer is correct in pointing out that the overall multiple comparison correction has to account for 6 comparisons. The code implementation of PDC used already accounts for the 2 directions of the synchronization testing, as this is integral to the PDC estimation. Therefore, our additional division of the threshold by 3 cumulatively resulted in accounting for all 6 comparisons.
Since Demetra’s session 15 CORE-OM was 10 it was not <10 as written in the text, and if the threshold for clinical significance is 10, then her score did not decrease to the non-clinical level. Right?
Thank you for your comment and suggestion. Demetra’s CORE-OM score indeed decreased to 10 points, which is the cut-off point for clinical distress and this sentence was revised to reflect this change accurately.
For better readability, please, increase the font of the title in figure 1 and possibly in figures 2 and 3 too.
Thank you for the suggestion, the font size of the title in figures 1, 2 and 3 is increased.
While in other windows the names of the clients are shown, in figure 4 clients are referred as C1 and C2 and therapist as T1. I would like to suggest that the line captures would be changed to include the names, e.g. C1 (Demetra). Then C1, C2 and T1 can be retained in the two panels.
Thank for the suggestion. The figure has now been revised accordingly.
Since the title of figure 4 is exactly the same as the figure caption, the title may not be necessary.
Thank for the suggestion. The figure has now been revised accordingly.
In references the titles of some references have sentence cases and some have each word capitalized. Please, check if the journal will accept this.
All references corrected and now have sentence cases
References 77 and 80 are not accurate enough.
Reference 77, now 79 corrected.
Reference 80, now 83 corrected.
Reviewer 3 Report
The study uses PDC as a novel approach to calculate psychophysiological synchrony in terms of hearth rate variability and examine its potential to contribute to our understanding of therapy process trough a single-case design (one couple therapy) in relation to the therapeutic alliance and the narrative analysis in the sessions. Only the third and penultimate sessions were analyzed in the perspesctive of syncronization / PDC.and narratives.
This is a frontier study that addresses to a very cogent topic in the scientific panorama. The most valuable aspect is the enhancement of idiographic design in this research vain, characterized by highly complexity, and it is very appreciable.
However, there are some areas to be improved:
First of all, the introduction and the discussion / conclusions section are too unbalanced. The introduction is very rich and extensive, well updated from the point of view of the literature, but it should be reorganized while maintaining a better, cleraler division between synchronization in general and synchronization in psychotherapy - if this is the structure that the authors intend to give to the introduction. In general perspective, attachment role on physiological syncronization is rilevent in this research niche and should be explicited at least in a couple of words (authors yet mentioned some of these studies in other sections of the manuscript, -citation 38 (Palmieri et al.) and 39 (Kleinbub et al.) - correctly referring to other aspects of these studies. To them, it should be added: Jaffe, J., Beebe, B., Feldstein, S., Crown, CL, Jasnow, MD, Rochat, P., & Stern, DN (2001) Rhythms of dialogue in infancy: Coordinated timing in development. Monographs of the society for research in child development, i-149). It is sufficient to add "...mental state related to attachment" in the list in lines 55-57.
Lines 76-78 should be improved. Authors wrote: "As such, several recent studies include psychophysiological measures in psychotherapy process research and treat physiological activation, and particularly its arousal component, as an index of affect [24]." but citation 24 is not recent, it is dated 1980. To this could be added, for example: Gennaro, A., Carola, V., Ottaviani, C., Pesca, C., Palmieri, A., & Salvatore, S . (2021). Affective Saturation Index: A Lexical Measure of Affect. Entropy, 23 (11), 1421. -which also considers cardiac parameters. Paragraph on alliance and syncronization should be shortened.
The final part is too little in-depth and not fully anchored to the theoretical premises - and some of these should be taken up and associated better with the interpretation of the results. For example, the apparently discordant results with respect to the general scientific trend conceiving IPS as a neural correlate of affiliative/empathic behaviors, should be explicitly re-examined in the conclusive secton to better interpret the results. In other words: it is a result in partial countertrend to most literature, and as such it should be better argued. A further interpretation of the significant lack of synchronization in the most salient sessions could consist in the fact that the intensity of the emotion led to a relational withdrawal because the patient needed a detachment from the context to concentrate on her own self-regulation, other than interpreting the asynchronization as a synonym of co-regulation. Beebe and Lachmann also specify that the amount of optimal syncronization during a prolonged interaction should be 30 percent of the time, beyond which self-regulation processes would be affected (see: Beebe, B., & Lachmann, FM (2013). Infant research and adult treatment: Co-constructing interactions. Routledge.). In a recent article the theories in contrast to seeing physiological synchronization just as as an affiliative phenomenon are outlined in detail, and a neural model is proposed to attempt a unifying explanation. The article in question is: Palmieri, A., Pick, E., Grossman-Giron, A., & Bitan, D. T. (2021). Oxytocin as the Neurobiological Basis of Synchronization: A Research Proposal in Psychotherapy Settings. Frontiers in Psychology, 12 . It is not necessary to quote this article, but the authors could find some interesting food for thought and further bibliographical references to expand the results interpretation in the discussion.
In the section of the method the data of the participants should be described in greater detail (age, educational level, etc. and also how long they have known each other, given that familiarity affects the ability to establish synchronization behaviors). As for the remaining part of the method, the statistical and methodological approach is adequate for the analysis of a single case.
Author Response
Response to Reviewer 3
The study uses PDC as a novel approach to calculate psychophysiological synchrony in terms of hearth rate variability and examine its potential to contribute to our understanding of therapy process trough a single-case design (one couple therapy) in relation to the therapeutic alliance and the narrative analysis in the sessions. Only the third and penultimate sessions were analyzed in the perspesctive of syncronization / PDC.and narratives.
This is a frontier study that addresses to a very cogent topic in the scientific panorama. The most valuable aspect is the enhancement of idiographic design in this research vain, characterized by highly complexity, and it is very appreciable.
However, there are some areas to be improved:
First of all, the introduction and the discussion / conclusions section are too unbalanced.
Thank you for this comment. Given that our focus in this paper centred on introducing PDC as a useful metric in the field of PS in psychotherapy, we recognize that we did not fully discuss the implications of the study’s findings. We have added several comments drawing upon the reviewer’s suggestions as detailed below.
The introduction is very rich and extensive, well updated from the point of view of the literature, but it should be reorganized while maintaining a better, cleraler division between synchronization in general and synchronization in psychotherapy - if this is the structure that the authors intend to give to the introduction.
Thank you for this comment. Although we recognize the usefulness of your suggestion to restructure this section of the introduction, we would like to keep the structure as it stands: we present key research on (behavioural) synchrony in psychotherapy and then literature on PS in psychotherapy; we do not review synchrony research in other interactional contexts, but only refer to such studies in order to illustrate the complexity and diversity of relevant findings to date. Drawing upon your comments, we have added reference to Palmieri et al (2021), which provides a very helpful conceptualization of the issues at hand.
In general perspective, attachment role on physiological syncronization is rilevent in this research niche and should be explicited at least in a couple of words (authors yet mentioned some of these studies in other sections of the manuscript, -citation 38 (Palmieri et al.) and 39 (Kleinbub et al.) - correctly referring to other aspects of these studies. To them, it should be added: Jaffe, J., Beebe, B., Feldstein, S., Crown, CL, Jasnow, MD, Rochat, P., & Stern, DN (2001) Rhythms of dialogue in infancy: Coordinated timing in development. Monographs of the society for research in child development, i-149). It is sufficient to add "...mental state related to attachment" in the list in lines 55-57.
Thank you for this comment. We agree that attachment security is a very important concept for the literature on PS -especially in the context of psychotherapy- and have now added a brief but more explicit reference to this as well as reference to the suggested studies.
Lines 76-78 should be improved. Authors wrote: "As such, several recent studies include psychophysiological measures in psychotherapy process research and treat physiological activation, and particularly its arousal component, as an index of affect [24]." but citation 24 is not recent, it is dated 1980. To this could be added, for example: Gennaro, A., Carola, V., Ottaviani, C., Pesca, C., Palmieri, A., & Salvatore, S . (2021). Affective Saturation Index: A Lexical Measure of Affect. Entropy, 23 (11), 1421. -which also considers cardiac parameters.
Thank you for this suggestion; the reference suggested has now been added.
Paragraph on alliance and syncronization should be shortened.
This paragraph has now been significantly shortened; it now includes reference only to the issues around the therapeutic alliance in couple therapy that are relevant to the findings in the study.
The final part is too little in-depth and not fully anchored to the theoretical premises - and some of these should be taken up and associated better with the interpretation of the results.
Thank you for this important comment. We have revised the Findings and Discussion section quite significantly and added further elaboration on possible interpretation of the key findings of our study. We have opted to add these primarily in the ‘Findings and Discussion’ rather than the ‘Conclusions’ section, in line with our preferred format for a mixed method single case study.
For example, the apparently discordant results with respect to the general scientific trend conceiving IPS as a neural correlate of affiliative/empathic behaviors, should be explicitly re-examined in the conclusive secton to better interpret the results. In other words: it is a result in partial countertrend to most literature, and as such it should be better argued. A further interpretation of the significant lack of synchronization in the most salient sessions could consist in the fact that the intensity of the emotion led to a relational withdrawal because the patient needed a detachment from the context to concentrate on her own self-regulation, other than interpreting the asynchronization as a synonym of co-regulation. Beebe and Lachmann also specify that the amount of optimal syncronization during a prolonged interaction should be 30 percent of the time, beyond which self-regulation processes would be affected (see: Beebe, B., & Lachmann, FM (2013). Infant research and adult treatment: Co-constructing interactions. Routledge.). In a recent article the theories in contrast to seeing physiological synchronization just as as an affiliative phenomenon are outlined in detail, and a neural model is proposed to attempt a unifying explanation. The article in question is: Palmieri, A., Pick, E., Grossman-Giron, A., & Bitan, D. T. (2021). Oxytocin as the Neurobiological Basis of Synchronization: A Research Proposal in Psychotherapy Settings. Frontiers in Psychology, 12 . It is not necessary to quote this article, but the authors could find some interesting food for thought and further bibliographical references to expand the results interpretation in the discussion.
Thank you for pointing us to the Palmieri et al. (2021) article, which does provide a very useful conceptualization of the complex correlates of PS, as well as for the thoughtful suggestions on possible interpretations re. the relational ‘meaning’ of shifts in PS observed in this single case. We have tried to develop the interpretation of these findings, although we recognize that any such interpretations need to be tentative given the single-case design of the study and its focus on introducing PDC in the field.
In the section of the method the data of the participants should be described in greater detail (age, educational level, etc. and also how long they have known each other, given that familiarity affects the ability to establish synchronization behaviors).
Thank you for your comment. We have now added the information on the couple suggested.
As for the remaining part of the method, the statistical and methodological approach is adequate for the analysis of a single case.

Round 2
Reviewer 2 Report
I accept the manuscript in the present form.
Reviewer 3 Report
I find the article much improved